# Pentraxin 3 in the cerebrospinal fluid during central nervous system infections: A retrospective cohort study

**Martin Munthe Thomsen**[1]\*, **Lea Munthe-Fog**[2], **Pelle Trier Petersen**[1], **Thore Hillig**[3], **Lennart Jan Friis-Hansen**[4], **Casper Roed**[1], **Zitta Barrella Harboe**[1,5], **Christian Thomas Brandt**[1,5,6]

**1** Department of Pulmonary and Infectious Diseases, Copenhagen University Hospital, Hillerød, North Zealand, Denmark, **2** Stemform/StemMedical, Cell Production and RnD, Søborg, Copenhagen Region, Denmark, **3** Department of Clinical Biochemistry, Copenhagen University Hospital, Hillerød, North Zealand, Denmark, **4** Department of Clinical Biochemistry, University Hospitals Bispebjerg and Frederiksberg, University of Copenhagen, Copenhagen, Denmark, **5** Department of Clinical Medicine, Faculty of Health and Medical Sciences, University of Copenhagen, Copenhagen Region, Copenhagen, Denmark, **6** Department of Infectious Diseases, Zealand University Hospital, University of Copenhagen, Roskilde, Denmark

\* martin.munthe.thomsen@regionh.dk

**Data Availability Statement:** The research data are pseudo-anonymized and contain potentially identifying patient information. According to Danish law these data can be shared with other

## Abstract

The present study describes diagnostic and prognostic abilities of Cerebrospinal fluid (CSF) Pentraxin 3 (PTX3) in central nervous system (CNS) infections. CSF PTX3 was measured retrospectively from 174 patients admitted under suspicion of CNS infection. Medians, ROC curves and Youdens index was calculated. CSF PTX3 was significantly higher among all CNS infections and undetectable in most of the patients in the control group, and significantly higher in bacterial infections compared to viral and Lyme infections. No association was found between CSF PTX3 and Glasgow Outcome Score. PTX3 in the CSF can distinguish bacterial infection from viral and Lyme infections and non-CNS infections. Highest levels were found in bacterial meningitis. No prognostic abilities were found.

## 1. Introduction

Infections of the central nervous system (CNS) cover a spectrum from self-limiting diseases with limited risk of sequelae to diseases with high morbidity, high risk of long-term sequelae, and high mortality [1, 2]. Encephalitis and bacterial meningitis remain among the most severe, with encephalitis case-fatality rates between 5–12% and sequelae in up to 60% of patients, and bacterial meningitis with a case fatality ranging from 14–21% and sequelae in a third of survivors [3–6]. Viral meningitis is less severe, but still with unfavorable outcomes of approximately 17%, although fatal cases are extremely rare [3].

The diagnosis of CNS infections is based on clinical presentation, cerebrospinal fluid (CSF) biochemistry, and microbiological analysis. Rapid diagnostics and therefore treatment has been shown to reduce mortality [7]. Despite relevant diagnostic measures, a pathogen is not identified in 30–40% of patients presenting with signs of CNS infections, and cases with the

qualified researchers after application to the Danish legal authorities, in this case the Regional Data Protection Agency, contact info: Forskningfortegnelse@regionsjaelland.dk. The Department of Infectious Diseases in Zealand region of Denmark can be contacted at: suh-ros-med@regionsjaelland.dk.

**Funding:** Receiver of grant: MMT Funder: Nordsjællands Hospital Url: https://www.nordsjaellandshospital.dk/forskning/Sider/default.aspx The funders had no role in study design, data collection and analysis, decision to publish, or preparation of the manuscript.

**Competing interests:** The authors have declared that no competing interests exist.

absence of pleocytosis occur [3, 8, 9]. There is yet no single acute phase response molecule that can reliably be used as a diagnostic discriminator between infectious pathogens or reflect the severity of the disease.

Pentraxins are part of the humoral innate immunity acting as antibody-like molecules, recognizing microbial moieties, possessing opsonic activity, and both activating and regulating the complement cascade as well as being involved in tissue remodeling [10, 11]. Pentraxin 3 (PTX3) is a long pentraxin molecule and has structural similarities to the well-known short pentraxins, C-reactive protein (CRP), and serum amyloid P (SAP). CRP and SAP are primarily produced in the liver, mainly by IL-6 induction in response to infection or inflammation, whereas PTX3 is mainly induced by IL-1 and TNF-$\alpha$ and produced by a variety of cells such as macrophages, endothelial cells, and dendritic cells and preserved in neutrophils for rapid distribution [10, 12, 13]. In laboratory settings, the PTX3 molecule remains stable despite several freeze-thaw cycles [13].

PTX3 is emerging as a marker for cardiovascular, inflammatory, and infectious diseases [14–20]. Serum levels of PTX3 have been described to be a strong marker of short-term overall mortality in hospitalized patients, in sepsis patients, and disease severity in critically ill patients. Furthermore, PTX3 has a low normal plasma range <2 ng/ml which can increase up to 1000 fold [21–23].

Knowledge of PTX3 inflammatory kinetics in the human brain during infection is lacking, but Zatta et al. [15] have shown that PTX3 in CSF increased during infection, with different levels in bacterial and viral CNS infections. Also, intracerebroventricular injection of lipopolysaccharide and IL-1 has shown PTX3 expression in mouse brain [24].

We aimed to evaluate the diagnostic and prognostic performance of CSF PTX3 in a variety of CNS infections in adult patients admitted at an Infectious Diseases department.

## 2. Methods

Adult ($\geq$ 18 years) patients admitted to the University Hospital North Zealand, in Copenhagen, Department of Infectious diseases between June 2016 and August 2019 clinically suspected of having meningitis, encephalitis, or Lyme neuroborreliosis were included. Nordsjællands Hospital services a population of 320.000 individuals. Patients were included prospectively in an observational study and samples were analyzed retrospectively.

CSF- and blood biochemistry results were obtained from the laboratory database LABKA II (*Dedalus Healthcare ApS, Denmark*). Blood biochemistry data obtained on the same day as lumbar puncture was performed were included. Data on microbiology was retrieved from databases for the individual departments.

Clinical data were collected from patient medical records including clinical datasheets, nurse registration files, and discharge records.

*Outcome* at discharge was categorized using the Glasgow Outcome Scale (GOS): (1) death; (2) vegetative state; (3) severe sequelae and dependency upon others in daily life; (4) moderate sequelae but with the ability to live independently; and (5) no or mild sequelae. An unfavorable outcome was defined as a GOS score of 1–4.

The included meningitis, encephalitis, or Lyme neuroborreliosis patients were $\geq$ 18 years of age and had a clinical appearance suggestive of CNS infection (any combination of neck stiffness, fever, headache or altered mental status or neurological symptoms) with $\geq$10 x $10^6$ cells/L in the CSF in combination with specific criteria as described below.

The control group was patients $\geq$ 18 years of age admitted under suspicion of CNS infection but with normal CSF biochemistry (CSF white blood cell count under 5 x $10^6$ cells/L, lactate under 2,4 mmol/L and protein < 0,8 g/L with no sign of blood contamination in the CSF) and with clinical improvement without specific meningitis treatment.

## 2.1 Lyme neuroborreliosis inclusion criteria

Presence of neurological symptoms in combination with a CSF leukocyte count $\geq 10 \times 10^6$ cells/L and positive intrathecal *B. burgdorferi* antibody production or 2) Presence of neurological symptoms raising the primary suspicion of borrelia in combination with CSF leucocytes $\geq 10 \times 10^6$ cells/L and positive blood *B. burgdorferi* antibody production

## 2.2 Bacterial meningitis inclusion criteria

Patients $\geq$ 18 years of age presenting with clinical disease suggesting bacterial meningitis (headache, fever, stiffness of the neck, petechiae, confusion or impaired level of consciousness) in combination with one or more of the following:

a.  Positive CSF culture.

b.  Positive blood culture and one or more of the following CSF findings: glucose index $<0.3$; CSF glucose $<2.0$ mmol/L or CSF lactate $>3.5$ mmol/L; protein $>2.0$ g/L.

c.  Presence of bacteria in gram stain of CSF or DNA/PCR identification of bacteria in the CSF.

Bacterial meningitis with unknown pathogen was included based on the clinical criteria above in combination with the following CSF biochemistry: $>10 \times 10^6$ cells/L) in combination with low CSF glucose or glucose-ratio ($<2.0$ mmol/L and 0.3 respectively) or CSF lactate $>3.5$ mmol /L.

## 2.3 Viral meningitis inclusion criteria

Clinical appearance suggestive of viral meningitis (eg headache, neck stiffness, fever, photophobia, GCS 14 or 15) and either of the following criteria:

1.  Positive viral DNA/RNA analysis of CSF

2.  Positive intrathecal antibody index for Herpes simplex virus (HSV) or Varicella Zoster virus (VZV)

3.  Serology suggestive of acute infection with a known CNS pathogen, (e.g. Tick-borne encephalitis virus)

4.  CSF leukocytes $>10 \times 10^6$ cells/L with mononuclear predominance and no other diagnosis considered more likely given all available information.

## 2.4 Encephalitis inclusion criteria

A clinical presentation suggestive of encephalitis (e.g. impaired consciousness $>24$ hours, headache, neurological deficit, seizures) and either of the following criteria:

1.  Positive viral DNA/RNA analysis of CSF

2.  Positive intrathecal antibody index for HSV/VZV

3.  CSF leukocytes $>10 \times 10^6$ cells/L and serology suggestive of acute infection with a known CNS pathogen,

4.  CSF leukocytes $>10 \times 10^6$ cells/L with mononuclear predominance and/or CNS imaging suggestive of encephalitis and no other diagnosis considered more likely given all available information.

## 2.5 Specimen storage and analysis

As part of standard care, an extra CSF vial was collected for supplemental diagnostic procedures. Extra vials were immediately stored at 4˚C and frozen at -20˚C within 24h and thereafter placed in storage -80˚C at patient discharge. CSF was thawed, mixed thoroughly, and spun 5 minutes at 4˚C, 1800 g and subsequently frozen at -80˚C in 500 μL aliquots in Matrix Cryotubes (#3744-WP1D, Thermo Scientific, Hudson, USA). CSF samples were thawed at 4˚C, mixed thoroughly, and measured in duplicate with an R-plex Pentraxin-3 assay on the MSD Quickplex 120 platform (Mesoscale Diagnostics, Maryland, USA) according to manufacturer instructions. Each sample was analyzed in duplicate with mean results reported. Non detectable concentrations of PTX3 (<3.2 pg/ml) was registered as 1 pg/ml. Duration of a PTX3 assay on 40 samples was approx. 3 hours.

## 2.6 Statistical analysis

Continuous data are presented as medians with an interquartile range (IQR) and categorical as n/N (%) Differences in continuous data between groups were compared using Kruskal-Wallis test with Dunn's post hoc tests. Receiver operating characteristics curves (ROC) were used to identify cut-off values for CSF PTX3, CSF lactate and CSF leukocyte cell count to distinguish between the different CNS infections and control group patients. Youden J statistic/index (J = Sensitivity + (Specificity—1)) was calculated to select a cut-off for maximizing classification accuracy. Sensitivity, specificity, positive predictive value (PPV), and negative predictive value (NPV) for the cut-off were calculated. SAS enterprise guide v. 7.1 and GraphPad Prism V. 9 were used for statistical analysis and graphics.

## 2.7 Ethics

The study was approved by the Danish Data Protection Agency (record no. 2012-58-0004 and 2012-58-0018). CSF storing of samples was approved (record no. 2013-41- 2502 and NOH-2017-029).

## 3. Results

A total of 174 patients were included. Demographics for 140 patients with a definite or probable CNS infection and 34 patients, where this was ruled out (control group), are presented in Tables 1 and 2.

A microbiological diagnosis was obtained in 116 patients (83%). A diagnosis of culture-negative bacterial meningitis was made in n = 6 patients (4%) and viral meningitis or encephalitis with the unknown pathogen in n = 12 (9%) and n = 6 (4%), respectively. Six patients were classified as miscellaneous (TB, HIV, syphilis, EBV myelitis, Influenza B, septic embolism from endocarditis). This group was not included in the data analysis.

Mortality among patients with bacterial meningitis including culture-negative cases was 7% (2 of 28 patients) and mortality among encephalitis patients 7% (1 of 15). One patient with HIV died and one elderly patient with neuroborreliosis died of sudden cardiac arrest of unknown cause.

Patients with a diagnosis of bacterial meningitis were significantly older (median age 66, IQR 51–75) compared to patients with viral meningitis (median age 36, IQR 27–52) (p<0.001), but not compared to other groups. Patients with a diagnosis of bacterial meningitis or viral encephalitis presented on admission with a significantly lower GCS compared to all other groups combined (p<0.001). Adverse outcome (GOS 1–4) was significantly more

**Table 1. Baseline characteristics of CNS infections with known aetiology.**

|  | All | Bacterial meningitis | Viral meningitis | Viral encephalitis | Neuro-borreliosis | Controls |
|---|---|---|---|---|---|---|
|  | N = 145 | N = 22 | N = 51 | N = 11 | N = 27 | N = 34 |
| Age | 52 (35–68) | 67 (50–76) | 35 (26–48) | 66 (45–71) | 61 (53–71) | 47 (36–66) |
| Male | 66/145 (46) | 13/22 (60) | 16/51 (31) | 7/11 (60) | 17/27 (63) | 13/34 (38) |
| Duration of symptoms (days) | 3 (1–8) | 3 (1–6) | 2 (1–4) | 4 (3–8) | 14 (3–23) | - |
| Diabetes | 7/106 (7) | 4/22 (18) | 2/48 (4) | 0/11 (0) | 1/25 (4) | - |
| Drug-induced immunosuppression | 6/106 (6) | 2/22 (9) | 1/48 (2) | 2/11 (18) | 1/25 (4) | - |
| GCS score | 15 (14–15) | 13 (11–15) | 15 (15–15) | 15 (13–15) | 15 (15–15) | - |
| Temperature (C) | 37.7 (37.2–38.5) | 38.5 (37.2–38.9) | 38.0 (37.5–38.8) | 38.1 (37.4–38.5) | 37.0 (36.8–37.6) | - |
| Headache | 62/92 (68) | 6/14 (43) | 47/48 (98) | 3/10 (30) | 3/20 (15) | - |
| Nausea | 34/75 (45) | 2/16 (13) | 26/46 (57) | 5/11 (46) | 0/2 (0) | - |
| Photo- or phonophobia | 32/60 (53) | 4/11 (36) | 27/45 (60) | 1/2 (50) | 0/2 (0) | - |
| Neck stiffness | 28/69 (41) | 9/19 (47) | 17/45 (38) | 1/3 (33) | 0/2 (0) | - |
| GOS score discharge |  |  |  |  |  |  |
| 1 | 4/102 (4) | 2/19 (11) | 0/48 (0) | 1/11 (9) | 1/24 (4) | - |
| 2 | 0/102 (0) | 0/19 (0) | 0/48 (0) | 0/11 (0) | 0/24 (0) | - |
| 3 | 6/102 (6) | 3/19 (16) | 0/48 (0) | 3/11 (27) | 0/24 (0) | - |
| 4 | 33/102 (33) | 7/19 (37) | 18/48 (38) | 1/11 (9) | 8/24 (33) | - |
| 5 | 58/102 (57) | 7/19 (37) | 30/48 (63) | 6/11 (55) | 15/24 (63) | - |
| C-reactive protein (mg/L) | 13 (2–79) | 221 (148–320) | 7 (2–17) | 8 (4–17) | 2 (2–2) | 39 (5–79) |
| B-leukocytes (x10⁹ cells/L) | 8.9 (6.9–13.1) | 17.2 (12.7–22.0) | 7.8 (6.5–9.9) | 8.7 (6.4–13.4) | 7.7 (6.4–9.3) | 9.9 (7.1–12.6) |
| CSF leukocytes (x10^6 cells/L) | 70 (4–305) | 2200 (928–7800) | 112 (25–299) | 120 (48–328) | 135 (20–238) | 2 (1–3) |
| CSF polynuclear (x10^6 cells/L) | 3 (0–45) | 1870 (569–7300) | 6 (1–31) | 6 (0–57) | 2 (0–5) | 0 (0–0) |
| CSF mononuclear (x10^6 cells/L) | 57 (4–207) | 159 (46–441) | 95 (27–263) | 71 (45–322) | 126 (23–245) | 2 (1–3) |
| CSF glucose (mmol/L) | 3.7 (3.3–4.1) | 2.7 (0.3–4.9) | 3.5 (3.3–3.9) | 3.8 (3.4–4.0) | 3.4 (2.9–4.2) | 4.0 (3.8–4.2) |
| CSF protein (g/L) | 0.70 (0.41–1.37) | 4.32 (0.96–5.03) | 0.68 (0.47–1.03) | 0.90 (0.68–1.46) | 1.20 (0.54–1.90) | 0.37 (0.29–0.47) |
| CSF lactate (mmol/L) | 2.2 (1.8–3.0) | 9.2 (4.1–14.6) | 2.3 (1.9–2.8) | 2.7 (2.0–3.6) | 2.1 (2.0–3.2) | 1.7 (1.5–2.0) |
| CSF PTX3 (pg/ml) | 6 (1–25) | 1010 (168–2560) | 8 (2–18) | 5 (2–25) | 7 (2–14) | <3,2 |

Data are shown as number and percentage or median and interquartile range.

common among patients with bacterial meningitis (62%) compared to patients with viral meningitis (29%, p<0.003) but not compared to encephalitis (53%, p>0.05).

## 3.1 CSF PTX3

PTX3 CSF concentrations are shown in Fig 1 and Tables 1–3.

Three patients with either very high or very low PTX3 CSF were included in the data analysis.

One patient with *Haemophilus influenzae* meningitis had non-detectable PTX3 in CSF despite CSF pleocytosis of 7870 x 10⁶ cells/L.

One patient with otitis media had a lumbar puncture performed with 13 cells in the CSF and high s-CRP, PTX3 <3,2 pg/ml, and surprisingly *S. pneumoniae* growth in CSF on day 3.

One patient with pneumococcal meningitis had very high CSF PTX3 at 26964 ng/ml.

**3.1.1 Bacterial meningitis.** Levels of CSF PTX3 concentrations among patients with culture-confirmed bacterial meningitis (1010, IQR 168–2560) were significantly higher than among patients with viral meningitis (8, IQR 2–18, p<0.0001), viral encephalitis (4, IQR 2–25, p<0.02), Lyme neuroborreliosis (7, IQR 2–14, p<0.0001), and controls (1, IQR 1–1, p<0.0001). When including culture-negative bacterial meningitis and viral meningitis with

**Table 2. Baseline characteristics of CNS infections with known and unknown etiology.**

| | Bacterial meningitis (known and unknown) | Viral meningitis (known and unknown) | Viral encephalitis (known and unknown) |
|---|---|---|---|
| | N = 28 | N = 63 | N = 16 |
| Age | 66 (51–75) | 36 (27–52) | 63 (50–69) |
| Male | 17/28 (61) | 22/63 (35) | 10/16 (63) |
| Duration of symptoms (days) | 2 (1–5) | 2 (1–3) | 4 (2–8) |
| Diabetes | 6/26 (23) | 2/55 (4) | 1/16 (6) |
| Drug-induced immunosuppression | 4/26 (15) | 1/55 (2) | 2/16 (13) |
| GCS score | 13 (11–15) | 15 (15–15) | 15 (14–15) |
| Temperature (C) | 38.5 (37.3–38.9) | 38.0 (37.5–38.6) | 37.9 (37.3–38.2) |
| Headache | 10/18 (56) | 52/54 (96) | 9/14 (64) |
| Nausea | 3/19 (16) | 30/53 (59) | 6/16 (38) |
| Photo- or phonophobia | 4/13 (31) | 31/51 (61) | 1/4 (25) |
| Neck stiffness | 11/23 (48) | 19/51 (37) | 3/6 (50) |
| GOS score discharge | | | |
| 1 | 2/23 (9) | 0/55 (0) | 1/15 (7) |
| 2 | 0/23 (0) | 0/55 (0) | 0/15 (0) |
| 3 | 4/23 (17) | 0/55 (0) | 4/15 (27) |
| 4 | 9/23 (39) | 19/55 (35) | 3/15 (20) |
| 5 | 8/23 (35) | 36/55 (66) | 7/15 (47) |
| C-reactive protein (mg/L) | 216 (121–320) | 7 (2–21) | 8 (2–19) |
| B-leukocytes (x10¨9 cells/L) | 16.0 (13.0–20.7) | 8.1 (6.4–10.5) | 10.4 (6.5–13.4) |
| CSF leukocytes (x10¨6 cells/L) | 1850(191–7420) | 65 (21–219) | 113 (47–219) |
| CSF polynuclear (x10¨6 cells/L) | 1680 (145–6770) | 7 (1–21) | 3 (1–45) |
| CSF mononuclear (x10¨6 cells/L) | 158 (46–466) | 57 (14–200) | 74 (43–159) |
| CSF glucose (mmol/L) | 2.8 (1.1–3.9) | 3.6 (3.3–4.1) | 3.8 (3.5–4.1) |
| CSF protein (g/L) | 2.79 (1.04–4.86) | 0.66 (0.44–1.01) | 0.90 (0.73–1.49) |
| CSF lactate (mmol/L) | 7.3 (4.1–12.8) | 2.3 (1.9–2.7) | 2.4 (2.0–3.5) |
| CSF PTX3 (pg/ml) | 653 (105–1871) | 7 (<3.2–18) | 7 (<3.2–20) |

Data are shown as number and percentage or median and interquartile range.

unknown pathogen, CSF PTX3 remained significantly increased among patients with bacterial meningitis compared to patients with viral meningitis (p<0.0001), viral encephalitis (p = 0.012), Lyme neuroborreliosis (p<0.0001), and controls (p<0.0001).

**3.1.2 Viral meningitis.** CSF PTX3 in patients with viral meningitis was significantly higher than among control group patients irrespectively of the inclusion of viral meningitis with an unknown pathogen (p = 0.0001).

**3.1.3 Viral encephalitis.** CSF PTX3 in patients with encephalitis (HSV-1 and 2, n = 5; VZV, n = 1; TBE, n = 4) were significantly higher than among control group patients also when including encephalitis with an unknown pathogen (p = 0.027 and p = 0.004, respectively).

**3.1.4 Lyme neuroborreliosis.** CSF PTX3 in patients with Lyme neuroborreliosis had significantly increased levels of PTX3 compared to the control group (p = 0.005).

**3.1.5 Other neuroinfectious.** Six patients were diagnosed with other infections and not included in the data analysis. Cerebral TB (n = 1, PTX3 <3.2 pg/ml), *S. mitis* endocarditis with septic embolies (n = 1, PTX3 4.9 pg/ml), neurosyphilis (n = 1, PTX3 <3.2 pg/ml), HIV (n = 1,

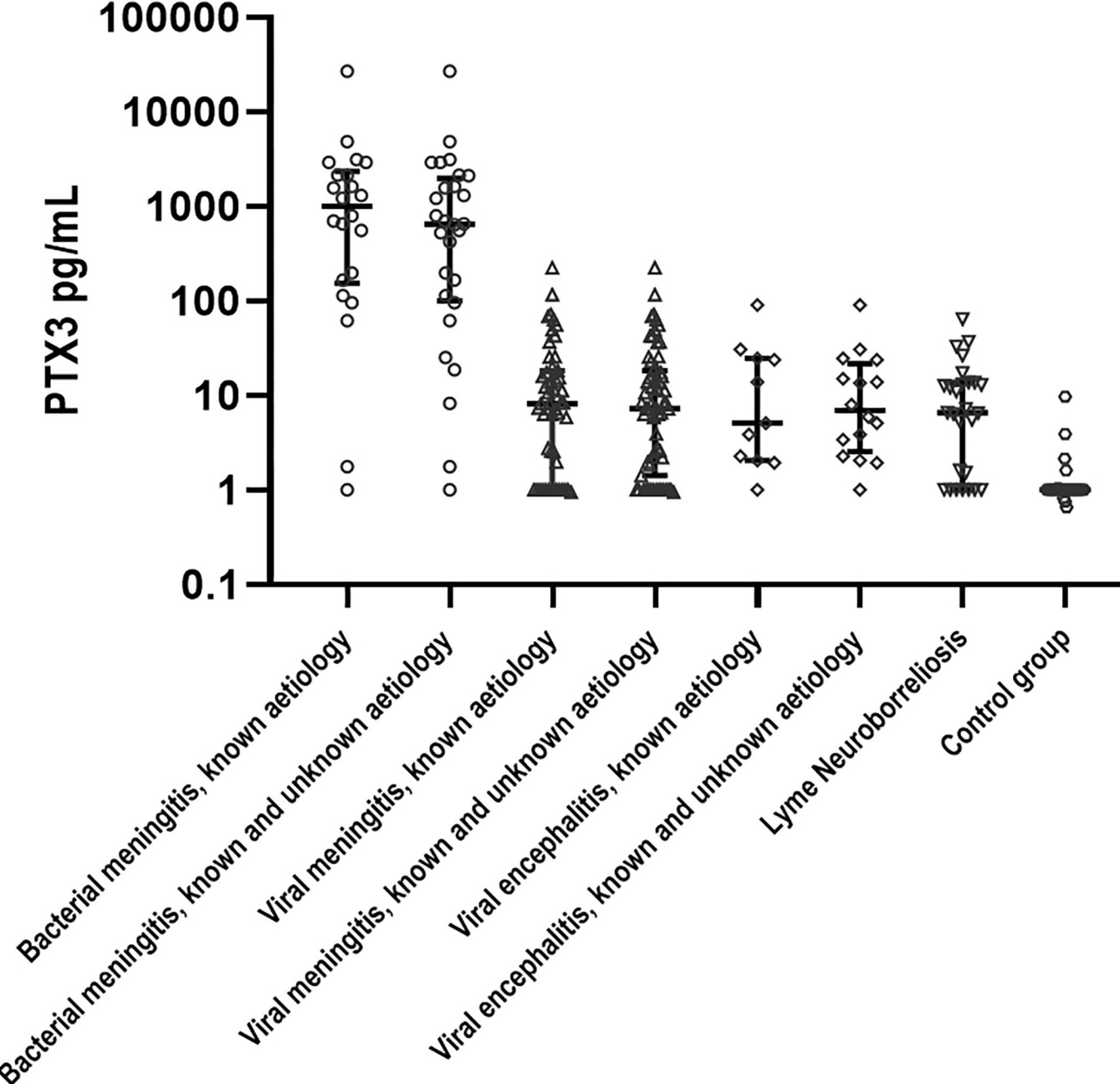

**Fig 1. Scatter dot plot of PTX3 concentration in the CSF.** Error bars indicate IQR.

PTX3 <3,2 pg/ml), influenza (n = 1, PTX3 <3.2 pg/ml) and EBV radiculitis (n = 1, PTX3 <3.2 pg/ml).

### 3.2 Diagnostic cut-off

Cut-off CSF concentrations of PTX3, lactate and leukocyte cell count and positive-and nega-tive predictive values (PPV and NPV) are shown in Fig 2 and Table 4.

Bacterial meningitis is distinguished from viral meningitis and encephalitis divided into microbiologically confirmed cases excluding and including culture-negative cases. Viral

**Table 3. PTX3 concentration in CSF based on pathogen.**

| | Median PTX3 (pg/ml) | | Median PTX3 (pg/ml) |
|---|---|---|---|
| Haemophilus Influenzae (n = 3) | 199.0 (<3.2–1626.5) | Enterovirus (n = 25) | 11.6 (6.3–40.0) |
| Listeria monocytogenes (n = 1) | 1219.0 | HSV 1+2 (n = 22) | 6.8 (<3.2–14.4) |
| Neisseria meningitidis (n = 3) | 1576.3 (660.0–2941.1) | VZV (n = 11) | <3.2 (IQR< 3.2) |
| S. anginosus (n = 1) | 2943,2 | | |
| S. aureus (n = 3) | 114.1 (96.5–559.8) | Lyme Neuroborreliosis (n = 27) | 6.6 (<3.2–13.7) |
| S. dysgalactiae (n = 3) | 802.8 (167.9–1315.3) | | |
| S. pneumoniae (n = 5) | 3148 (1081.1–15929.2) | | |
| Streptococcus bovis (n = 1) | 700.9 | | |
| Streptococcus mitis (n = 1) | 2116.2 | | |

Data shown as median and IQR.

meningitis, encephalitis, and Lyme neuroborreliosis is distinguished from control groups also divided into microbiologically confirmed cases excluding and including culture-negative cases.

When comparing the diagnostic capability of CSF PTX3 and CSF lactate, CSF PTX3 had higher PPV and NPV on all compared diagnoses except when distinguishing encephalitis of known and unknown etiology from the control patients where the NPV of CSF lactate was highest.

When comparing the diagnostic capability of CSF PTX3 and CSF leukocyte cell count the PPV and NPV for distinguishing bacterial meningitis from viral meningitis and encephalitis was similar, although CSF PTX3 performed slightly better when bacterial meningitis of unknown etiology were included. CSF leukocyte cell count performed better when distinguishing between viral meningitis, encephalitis and Lyme neuroborreliosis.

### 3.3 CSF levels of PTX3 and outcome

Comparing levels of PTX3 in the CSF between patients with adverse outcomes (GOS 1–4) to good outcomes (GOS 5) is shown in Table 5. The results yielded no significant differences when comparing within individual disease groups, ($p > 0.05$).

### 3.4 CSF biochemistry and PTX3

CSF PTX3 were closely correlated to pooled data for CSF cell count, CSF polynuclear cell count (PNC), and CSF protein concentration ($p < 0.0001$, Spearman rank respectively 0.75, 0.68, and 0.61). See Fig 3.

## 4. Discussion

To our knowledge, this is the first study using the Pentraxin-3 assay on the MSD Quickplex 120 platform showing a near to zero pg/ml concentration of PTX3 in CSF in a control group and concurrent highly elevated CSF PTX3 in patients with bacterial meningitis, viral meningitis, and viral encephalitis.

We found, that PTX3 was significantly higher in the CSF of patients admitted with culture-confirmed bacterial meningitis compared to those with viral meningitis, viral encephalitis, neuroborreliosis, and a control group of patients without CNS infection. Our results support the hypothesis that PTX3 can be used as a biomarker to discriminate bacterial meningitis from viral infections and is a promising clinical decision tool to rule out bacterial meningitis.

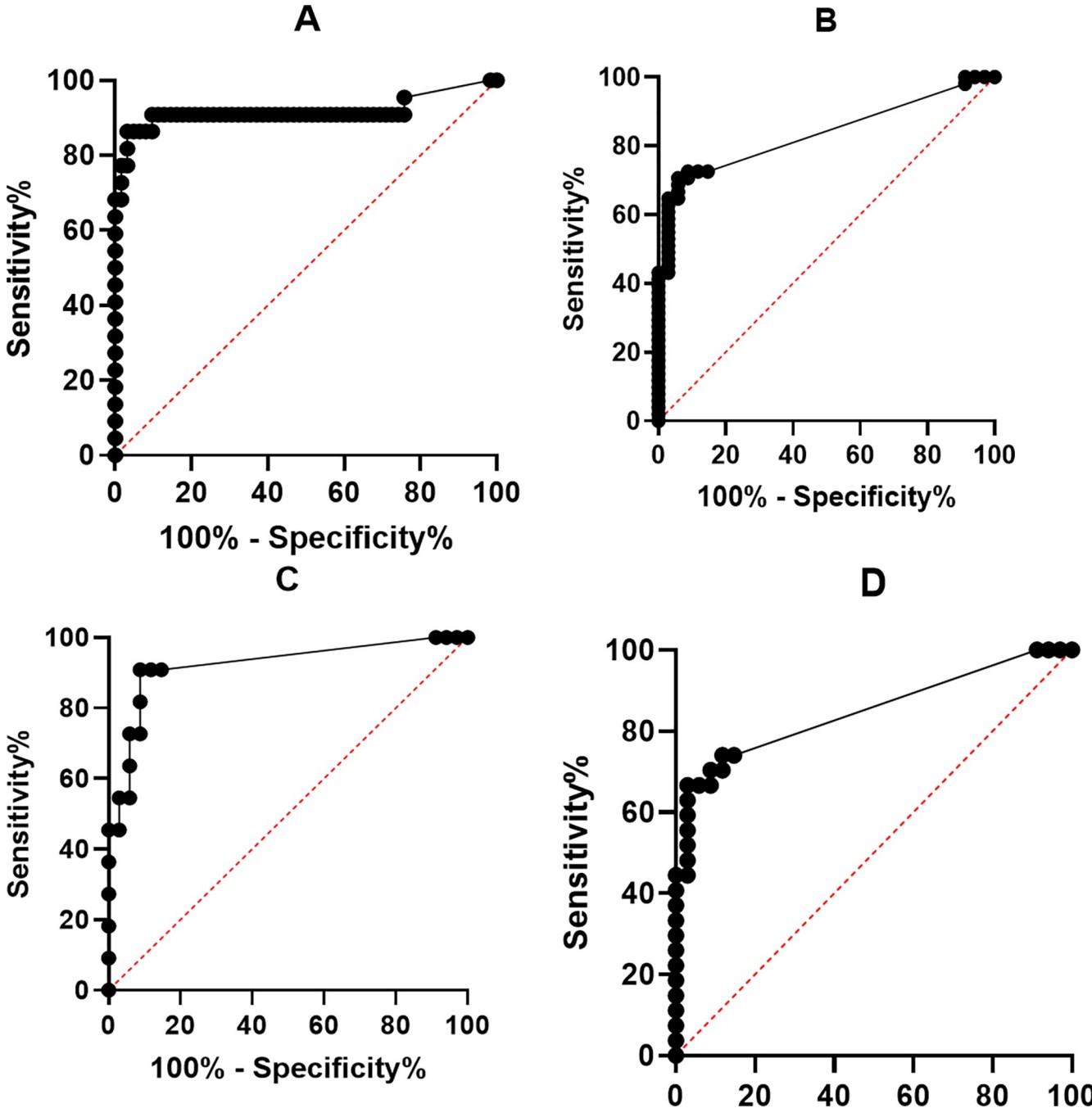

**Fig 2. ROC curves for PTX3 diagnostic abilities.** A) Bacterial meningitis versus Viral meningitis and Viral encephalitis. B) Viral meningitis versus Control group. C) Viral encephalitis versus Control group. D) Lyme disease versus Control group.

In comparison with studies of other biomarkers, we find PPV and NPV for CSF PTX3 comparable to known used biomarkers, including CSF cell count, CSF polynuclear cell count (PNC), and CSF protein concentration. We found PTX3 to be non-detectable or very low in the CSF in a control group despite clinical suspicion of meningitis and other inflammatory activity as indicated by elevated CRP and leukocytes in the blood. Our findings suggest that PTX3 does not cross the blood-brain barrier in patients without inflamed meninges, albeit we

**Table 4. Table of AUC, Youdens J, sensitivity, specificity and predictive values.** A) PTX3, B) lactate and C) leukocyte cell count in the CSF for diagnosing CNS infections.

**A)**

| Microbiologically confirmed pathogen | AUC | Maximum Youdens J (pg/ml) | Sensitivity | Specificity | PPV | NPV |
|---|---|---|---|---|---|---|
| Bacterial vs. viral meningitis and encephalitis, known aetiology | 0.917 (95% CI 0.819 to 1.000, p<0.0001) | 93.5 | 86.4% | 96.8% | 91% | 95% |
| Viral known aetiology vs. controls | 0.836 (95% CI 0.751–0.921, p<0.0001) | <3.2 | 70.6% | 94.1% | 95% | 68% |
| Encephalitis known aetiology vs. controls | 0.923 (95% CI 0.818–1.000, p<0.0001) | <3.2 | 90.9% | 91.2% | 77% | 97% |
| Lyme neuroborreliosis vs controls | 0.849 (95% CI 0.744–0.9528, p<0.0001) | 4.62 | 66.7% | 97.1% | 95% | 79% |
| **Microbiologically confirmed pathogen and culture negative samples** | | | | | | |
| Bacterial known+unknown aetiology vs. Viral and encephalitis known and unknown | 0.903 (CI 95% 0.817–0.989, p<0.0001) | 93.5 | 78.6% | 97.4% | 92% | 91% |
| Viral known+unknown aetiology vs. Controls | 0.852 (95% CI 0.777–0.927, p<0.0001) | <3.2 | 74.6% | 91.2% | 94% | 66% |
| Encephalitis known+unknown aetiology vs. controls | 0.939 (95% CI 0.862–1.000, p<0.0001) | <3.2 | 93.80% | 91.2% | 83% | 97% |

**B)**

| Microbiologically confirmed pathogen | AUC | Maximum Youdens J (mmol/L) | Sensitivity | Specificity | PPV | NPV |
|---|---|---|---|---|---|---|
| Bacterial vs. viral meningitis and encephalitis, known aetiology | 0.919 (95% CI 0.819 to 1.000), p<0.0001 | 3.1 | 89.5% | 85.5% | 68% | 96% |
| Viral known aetiology vs. controls | 0,847 (95% CI 0,7615 to 0,9318), p<0.0001 | 2.2 | 66.0% | 89.7% | 91% | 62% |
| Encephalitis known aetiology vs. controls | 0,894 (95% CI 0.782 to 1.000), p = 0.0007 | 1.9 | 100.0% | 65.5% | 44% | 100% |
| Lyme neuroborreliosis vs controls | 0,830 (95% CI 0,714 to 0,946), p = 0.0001 | 2.1 | 68.4% | 82.8% | 74% | 92% |
| **Microbiologically confirmed and culture negative patient groups** | AUC | Maximum Youdens J (mmol/L) | Sensitivity | Specificity | PPV | NPV |
| Bacterial known+unknown aetiology vs. Viral and encephalitis known and unknown | 0.932 (95% CI 0.851 to 1.000), p<0.0001 | 3.1 | 91.3% | 85.3% | 68% | 97% |
| Viral known+unknown aetiology vs. Controls | 0,809 (95% CI 0,719 to 0,899), p<0.0001 | 2.2 | 61.4% | 89.7% | 92% | 54% |
| Encephalitis known+unknown aetiology vs. controls | 0,903 (95% CI 0,809 to 0,997), p<0.0001 | 1.9 | 100.00% | 65.5% | 52% | 100% |

**C)**

| Microbiologically confirmed pathogen | AUC | Maximum Youdens J (x10^6/L) | Sensitivity | Specificity | PPV | NPV |
|---|---|---|---|---|---|---|
| Bacterial vs. viral meningitis and encephalitis, known aetiology | 0,860 (95% CI 0,7447 to 0,9758), p<0.0001 | 906 | 76.2% | 98.4% | 94% | 92% |
| Viral known aetiology vs. controls | 0,956 (95% CI 0,912 to 0,999), p<0.0001 | 7 | 90.0% | 100.0% | 100% | 73% |
| Encephalitis known aetiology vs. controls | 0,989 (95% CI 0,9647 to 1,000), p<0.0001 | 25 | 90.9% | 100.0% | 100% | 97% |
| Lyme neuroborreliosis vs controls | 0,963 (95% CI 0,9003 to 1,000), p<0.0001 | 6 | 91.7% | 100.0% | 100% | 94% |
| **Microbiologically confirmed and culture negative patient groups** | AUC | Maximum Youdens J (x10^6/L) | Sensitivity | Specificity | PPV | NPV |
| Bacterial known+unknown aetiology vs. Viral and encephalitis known and unknown | 0,842 (95% CI 0,7433 to 0,9404), p<0.0001 | 778 | 67.9% | 97.4% | 91% | 89% |

*(Continued)*

**Table 4.** (Continued)

**A)**

| Microbiologically confirmed pathogen | AUC | Maximum Youdens J (pg/ml) | Sensitivity | Specificity | PPV | NPV |
|---|---|---|---|---|---|---|
| Viral known+unknown aetiology vs. Controls | 0,957 (95% CI 0,918 to 0,996), p<0.0001 | 6 | 90.2% | 100.0% | 100% | 85% |
| Encephalitis known+unknown aetiology vs. controls | 0,993 (95% CI 0,976 to 1,00), p<0.0001 | 10 | 93.8% | 100.0% | 100% | 97% |

cannot conclude whether the PTX3 is produced intrathecally in patients with confirmed meningitis or passing from the blood through a permeable blood-brain barrier.

Studies have investigated PTX3 levels in CSF under different conditions [14, 15, 25], with PTX3 levels in sick and baseline patients higher than our findings using units ng/ml instead of our pg/ml. These findings do not correspond to our lower levels of PTX3 and are explained by different ELISA assays. Only a few others have studied CSF-PTX3 in humans during inflammation or infection. Lui et al. [25], find elevated PTX3 in anti-NMDAR encephalitis but without the ability to distinguish the group from a control group. A study by Zatta et al. [15], which included 19 patients diagnosed with bacterial or aseptic meningitis found results similar to ours with significantly higher CSF concentrations of PTX3 in bacterial CNS infections. The CSF findings correspond to what is also seen in plasma in sepsis patients [23].

Recognizing and rapidly diagnosing CNS infections in traumatic brain injury patients with extraventricular drainage is challenging due to symptomatology and elevated known CSF biomarkers affected by trauma itself [26], and analysis of CSF PTX3 in such a group of patients would be interesting.

Alons et al. [27], investigated CSF procalcitonin in patients with community-acquired bacterial meningitis and post neurosurgical intervention meningitis compared with CSF from patients not suspected for meningitis and lumbar punctured for other noninfectious reasons and found a ROC AUC 0,93, sensitivity 92%, specificity 68%, PPV 73%, and NPV 90%. The predictive values and AUC are lower than our findings for PTX3.

Buch et al. [28], investigated the known clinically used biomarkers (CSF leukocytes, CSF neutrophil fraction, CSF protein, CSF glucose ratio, Plasma-CRP, and CSF lactate) for distinguishing between acute bacterial meningitis and acute viral meningitis/encephalitis and found that CSF lactate with ROC AUC 0,976, sensitivity 96%, specificity 85%, PPV 72% and NPV 98% performed better than other CSF biochemistry. CSF-lactate had a higher sensitivity and NPV, but lower specificity and PPV than PTX3.

Egelund et al. [29], investigated PPV and NPV of CSF pleocytosis as a discriminator between bacterial meningitis and other brain infections and found PPV of 62% and NPV of 96%. Despite the close correlation between CSF pleocytosis and PTX3, this was inferior to our findings. However, the patients included in the study by Egelund et al. were more diverse also including brain abscesses.

In this cohort, we found PPV and NPV for CSF PTX3 as a marker of CNS infection better than CSF lactate for all investigated infectious diagnoses, and PPV and NPV equal to CSF leukocytes as a marker for bacterial meningitis, with a slightly better performance when including

**Table 5. PTX3 median values at different Glasgow Outcome Scores.**

| | Bacterial meningitis | Viral meningitis | Lyme neuroborreliosis | Viral encephalitis |
|---|---|---|---|---|
| GOS1-4 | 1267 (IQR 96–2943) | 8(IQR <3,2–15) | 11(IQR <3,2–35) | 4(IQR <3,2–57) |
| GOS 5 | 803 (IQR 660–2116) | 8(IQR <3,2–23) | 7(IQR<3,2–13) | 9(IQR <3,2–28) |

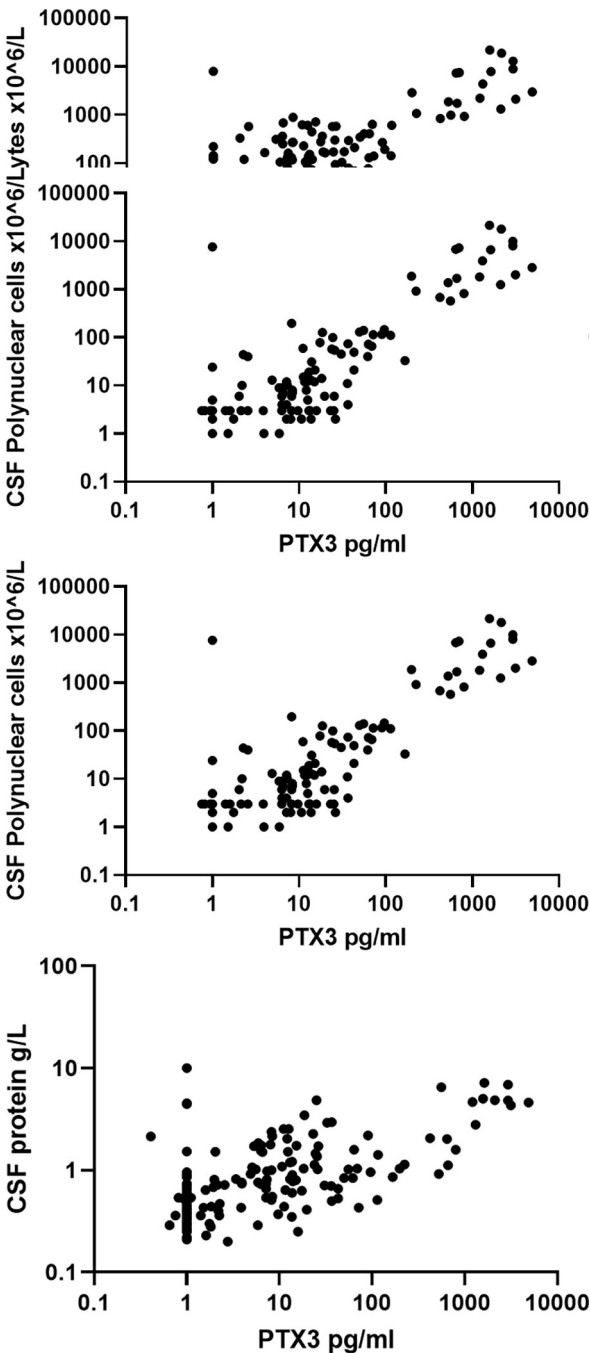

**Fig 3. PTX3 correlation to CSF cell count, CSF polynuclear cell count and CSF protein concentration.**

also infections of unknown etiology. CSF leukocyte cell counts performs better when distinguishing viral, encephalitis and Lyme neuroborreliosis from healthy controls.

The population in this study is primarily an urban population in a developed country with widespread vaccine coverage. During the last 50 years, improved prevention and introduction of first Haemophilus influenzae vaccine and later pneumococcal vaccines has changed the epidemiology of bacterial meningitis from being a disease primarily in children, to a disease occurring more commonly among the elderly [2, 3, 30]. In developing countries and countries

with high prevalence of meningococcal infections, bacterial meningitis is still most prevalent in children and young adults [31–33]. Proposing that the host immunological response is more potent in a young compared to an elderly population, this could result in a higher PTX3 response. An increased distance to a treatment facility in rural areas is likely to increase disease severity on presentation and this could also result in higher levels of PTX3, thereby increasing the predictive values of PTX3 as a marker for bacterial meningitis in these settings.

Previous studies have indicated that Plasma-PTX3 is associated with death and poor outcomes during hospital admissions and infections [21–23]. In our study we could not detect any association to a higher risk of death and poor outcome. This is explained by the limited number of patients in our study with poor outcomes.

The close correlation to CSF leukocytosis in especially bacterial meningitis suggests a limited prognostic value of PTX3 since CSF pleocytosis performs poorly as a prognostic factor and that primarily the absence of CSF pleocytosis is associated with poor outcomes. CSF polymorphonuclear cells (PNC) are useful biomarkers for bacterial infection, although cases of bacterial meningitis without CSF pleocytosis occur. Our finding of a correlation between CSF PNC and PTX3 is in agreement with previous findings by Jaillon et al. [12] who found PTX3 storage in neutrophil granules.

The strengths of the study is a relatively high number of unselected meningitis cases included and the complete follow-up of patients.

However, this is a single-center study using the same laboratory without concurrent patient inclusion and analysis performed at other centers. We were not able to control for the length of symptoms before hospital admission, a well-known factor associated with the level of inflammation at the time for lumbar puncture, which may affect the levels of PTX3. Also, some patients may have received antibiotics before the procedure. We did not measure parallel serum PTX3 concentrations, and thereby cannot exclude a possible spillover through the compromised blood-brain barrier to the CSF.

The freeze-thawing procedure of CSF may be less sensitive than analysis performed immediately on fresh samples.

## 5. Conclusion

The present study has described the possible value of the IL-1 and TNF- α driven PTX3 as a CSF biomarker for infections in the CNS. The study show that patients with bacterial meningitis presented with very high concentrations of PTX3 in the CSF on admission. Furthermore, CSF PTX3 could differentiate bacterial meningitis from other CNS infections and patients without meningitis. PTX3 may be used to identify patients with bacterial meningitis independently of prior treatment using PTX3 as a single biomarker.

CSF PTX3 in viral meningitis, encephalitis, and Lyme neuroborreliosis could only to a limited extent distinguish these infections from patients without a CNS infection.

PTX3 cut-off values with a high PPV and NPV was equal to or better than presently used CSF biomarkers when comparing other studies, and performs better than lactate in our cohort and similar to or slightly better than CSF leukocyte cell count when diagnosing bacterial meningitis. Immediate analysis of PTX3 on freshly sampled blood and CSF performed in 2 or more centers is necessary to document validity of this analysis prior to any routine inclusion.

## Supporting information

**S1 Table. Table of individual included patients CNS infectious pathogen and PTX3 concentration in the CSF.**
(XLSX)

## Author Contributions

**Conceptualization:** Martin Munthe Thomsen, Lennart Jan Friis-Hansen, Christian Thomas Brandt.

**Data curation:** Martin Munthe Thomsen, Pelle Trier Petersen, Thore Hillig.

**Funding acquisition:** Martin Munthe Thomsen, Christian Thomas Brandt.

**Investigation:** Martin Munthe Thomsen.

**Methodology:** Lea Munthe-Fog, Christian Thomas Brandt.

**Project administration:** Christian Thomas Brandt.

**Software:** Thore Hillig.

**Supervision:** Christian Thomas Brandt.

**Writing – original draft:** Martin Munthe Thomsen.

**Writing – review & editing:** Martin Munthe Thomsen, Lea Munthe-Fog, Pelle Trier Petersen, Thore Hillig, Lennart Jan Friis-Hansen, Casper Roed, Zitta Barrella Harboe, Christian Thomas Brandt.

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
