## [Decision Letter · Decision Letter 0]

27 Sep 2022

PONE-D-22-24326Pentraxin 3 in the cerebrospinal fluid during central nervous system infections: A retrospective cohort study.PLOS ONE

Dear Dr. Thomsen,

Thank you for submitting your manuscript to PLOS ONE. After careful consideration, we feel that it has merit but does not fully meet PLOS ONE’s publication criteria as it currently stands. Therefore, we invite you to submit a revised version of the manuscript that addresses the points raised during the review process.

We look forward to receiving your revised manuscript.

Kind regards,

Francesco Lolli, M.D., Ph.D.

Academic Editor

PLOS ONE

Journal Requirements:

Reviewers' comments:

Reviewer's Responses to Questions

**Comments to the Author**

1. Is the manuscript technically sound, and do the data support the conclusions?

Reviewer #1: Yes

Reviewer #2: Yes

2. Has the statistical analysis been performed appropriately and rigorously? 

Reviewer #1: Yes

Reviewer #2: Yes

3. Have the authors made all data underlying the findings in their manuscript fully available?

Reviewer #1: No

Reviewer #2: No

4. Is the manuscript presented in an intelligible fashion and written in standard English?

Reviewer #1: Yes

Reviewer #2: Yes

5. Review Comments to the Author

Reviewer #1: This is a well-designed and carefully conducted study with acknowledged limitations. Control group is very well selected to match real world clinical situation. I have no major criticisms and would like to suggest correction of a few typographical errors and imprecisions in the text, and optionally to provide some additional information as specified below:

ABSTRACT: In the first page (PLOS ONE Table), "random operator curves" are mentioned - I suppose rather "receiver operating characteristic" is correct. In page 3 of the Manuscript itself, the correct abbreviation ROC is used instead.

3rd paragraph: "CSF PTX3 was ... undetectable in the control group." 4 out of 34 patients of the control group apparently had detectable CSF PTX3 concentrations (see Fig. 1), so I would recomment to change the text accordingly (e.g. "... and undetectable in 30/34 patients in the control group" or "... and undetectable in most of patients in the control group" or otherwise).

Section 2.1 For practical reasons, the readers might be interested how long the CSF PTX-3 analysis takes.

Section 2.2 Statistics: page 8, row 203: the typographic error in the formula for Youden index should be corrected (missing "-", i.e., "J = Sensitivity + Specificity - 1" is correct, not "... Specificity 1").

Discussion section: page 12, row 305: missing space between "patient" and "group"

p. 12, row 313 and 323: the abbreviation P-CRP or P-PTX3 is not explained; perhaps it is clear for most readers that "P" stands for "plasma" but please consider to explain before the first use

p. 13, row 331: "neutrophil" instead of "neutrophile"

Table 1, 1st column: perhaps it is not necessary to repeat "(IQR) if it is already stated in the Table Title/Legend; row "GCS score": symbol "(" possibly introduced by mistake; row "CSF mono": I would prefer the term "mononuclear" (or "mononuclears" or "mononuclear cells") since "mono" may be interpreted either as "mononuclear" or "monocyte" (the same applies to Table 2)

Table 1, column "Neuroborreliosis": it is quite surprising that 1/24 patients died; was the death causally related to neuroborreliosis? (response optional since it is not of significant importance for the Manuscript)

Table 3: "Haemophilus influenzae" instead of "Haemophilus influenza"; "Neuroborreliosis" instead of "Neuroorreliosis"

Table 4: cut-off values below the detection limit of the assay are of dubious importance. Am I correct if I suppose that using the LoD stated (3.2 pg/ml) as a cut-off value would result in the same sensitivity, specificity, PPV and NPV? If I understand the Method section well, no values between 1.0 pg/ml (arbitrary value for samples with undetectable PTX-3 concentration) and 3.2 pg/ml (LoD of the assay) should exist.

Figure 3 on the right, y-axis: "CSF Polynuclear cells 10^6/L" not "...10^16/L"

References:

page 23, row 489:Reference 28 is incomplete.

It would be interesting to compare sensitivity, specificity, PPV and NPV of PTX-3 and other CSF biomarkers within the same patient cohort. I would like to bring the Authors´ attention to an excellent study of prof. T. O. Kleine et al. "New and old diagnostic markers of meningitis in cerebrospinal fluid (CSF)" (Brain Res Bull 2003; 61(3): 287-297) where it is stated that "tests with new markers were more laborious, expensive and time-consuming than CSF lactate test" (as well as other classical CSF tests). The same seems true after 20 years, and until now, none of the "new" markers studied in 2003 entered routine practice of CSF laboratories, at least not in the emergency service setting. It seems to me that with PTX-3 measured by an expensive and probably also time-consuming assay would also be quite impracticle, although such pilot studies are always welcomed and may possibly also bring new insights into disease pathophysiology.

Reviewer #2: The author needs to comment further on their case populations to other studies, especially from other areas, to compare the possible relevance of the test in other areas concerning severity, local origin and clinical case differential conditions.

Although other biomarkers are discussed, the results as compared to the other test already performed (i.e. CSF analysis reported, CSF cells and differential counts) can be compared in a diagnostic table similar to table 4 and fig3.

minor: a short title should be short

6. PLOS authors have the option to publish the peer review history of their article (what does this mean?). If published, this will include your full peer review and any attached files.

Reviewer #1: No

Reviewer #2: No

---

## [Author Response · Author response to Decision Letter 0]

13 Jan 2023

To the Editor 

PLOS ONE 

Response to Reviewers

Enclosed below our response to the editorial and reviewers’ comments.

Thank you for providing us with comments and criticism and allowing us to revise accordingly. The comments have been corrected or added in the best way without disagreements.

Journal Requirements:

Author response: Manuscript has been adjusted to the PLOS ONE’s style requirements. 

Author response: Due to legal restrictions in Denmark, sharing of pseudo anonymized data is only allowed after application from a qualified researcher to the Danish health authorities. We have provided a dataset with CSF PTX3 values and specific diagnoses and pathogens. A dataset with this information has been added to the submission (Supporting information; “S6 Table. Table of individual included patients CNS infectious pathogen and PTX3 concentration in the CSF”). 

Author response: We have corrected accordingly and removed the sentence “although we find no correlation between CSF protein and PTX3 (data not shown).” 

Author response: The incomplete reference 28 has been corrected. Reference style is changed to “Vancouver“ style. Reference 30-33 has been added as a response to reviewer’s comments.

Reviewers comments:

ABSTRACT: In the first page (PLOS ONE Table), "random operator curves" are mentioned - I suppose rather "receiver operating characteristic" is correct. In page 3 of the Manuscript itself, the correct abbreviation ROC is used instead.

Author response: “random operator curves” has been corrected to “receiver operating characteristic”. 

3rd paragraph: "CSF PTX3 was ... undetectable in the control group." 4 out of 34 patients of the control group apparently had detectable CSF PTX3 concentrations (see Fig. 1), so I would recommend to change the text accordingly (e.g. "... and undetectable in 30/34 patients in the control group" or "... and undetectable in most of patients in the control group" or otherwise).

Author response: 3rd paragraph: The text has been changed to “and undetectable in most of the patients in the control group” as suggested. And on p. 17 line 305 “We found PTX3 to be non-detectable or very low in the CSF…”

Section 2.1 For practical reasons, the readers might be interested how long the CSF PTX-3 analysis takes.

Author response: Section 2.1 on lines 149-150: The text “Duration of a PTX3 assay on 40 samples was approx. 3 hours” has been added.

Section 2.2 Statistics: page 8, row 203: the typographic error in the formula for Youden index should be corrected (missing "-", i.e., "J = Sensitivity + Specificity - 1" is correct, not "... Specificity 1").

Author response: Section 2.2, line 157: The missing “- “ has been added; “(J=Sensitivity + (Specificity – 1)”

Discussion section: page 12, row 305: missing space between "patient" and "group"

Author response: Discussion section page 16, row 295: Sentence has been corrected to “in such a group of patients”.

p. 12, row 313 and 323: the abbreviation P-CRP or P-PTX3 is not explained; perhaps it is clear for most readers that "P" stands for "plasma" but please consider to explain before the first use.

Author response: P. 16, row 302 and p. 17 row 328: The abbreviation has been corrected to “plasma-CRP and “plasma-PTX3”.

p. 13, row 331: "neutrophil" instead of "neutrophile"

Author response: p. 18, row 337: Typographical error has been corrected to ”neutrophil”.

Table 1, 1st column: perhaps it is not necessary to repeat "(IQR) if it is already stated in the Table 

Author response: Table 1, 1st column “IQR” and (%) has been deleted, as the they are already stated in the Table.

Title/Legend; row "GCS score": symbol "(" possibly introduced by mistake; row "CSF mono": I would prefer the term "mononuclear" (or "mononuclears" or "mononuclear cells") since "mono" may be interpreted either as "mononuclear" or "monocyte" (the same applies to Table 2)

Author response: Table 1 and Table 2: Symbol “(“ has been deleted and “CSF mono” has been corrected to “CSF mononuclear”

Table 1, column "Neuroborreliosis": it is quite surprising that 1/24 patients died; was the death causally related to neuroborreliosis? (response optional since it is not of significant importance for the Manuscript)

Author response: Section 3.0, page 9 lines 182 to 183: The text “and one elderly patient with neuroborreliosis died of sudden cardiac arrest of unknown cause.” Has been added.

Table 3: "Haemophilus influenzae" instead of "Haemophilus influenza"; "Neuroborreliosis" instead of "Neuroorreliosis"

Author response: Table 3: Typographical errors has been corrected as pointed out by the reviewer. 

Table 4: cut-off values below the detection limit of the assay are of dubious importance. Am I correct if I suppose that using the LoD stated (3.2 pg/ml) as a cut-off value would result in the same sensitivity, specificity, PPV and NPV? If I understand the Method section well, no values between 1.0 pg/ml (arbitrary value for samples with undetectable PTX-3 concentration) and 3.2 pg/ml (LoD of the assay) should exist.

Author response: Table 4: It is correct that values below 3.2 pg/ml was set to 1 pg/ml, and therefore no values between 1.0 pg/ml and 3.2 pg/ml should exist. The error has been corrected. It does not affect the sensitivity, specificity, NPV or PPV.

The same mistake has been corrected in Table 1 and 2. 

Figure 3 on the right, y-axis: "CSF Polynuclear cells 10^6/L" not "...10^16/L"

Author response: Figure 3: The error has been corrected as suggested. 

References:

page 23, row 489: Reference 28 is incomplete.

Page 21, row 448: The reference has been corrected to: 

“Buch K, Bodilsen J, Knudsen A, Larsen L, Helweg-Larsen J, Storgaard M, et al. Cerebrospinal fluid lactate as a marker to differentiate between community-acquired acute bacterial meningitis and aseptic meningitis/encephalitis in adults: a Danish prospective observational cohort study. Infect Dis. 2018 Jul 3;50(7):514–21.“

It would be interesting to compare sensitivity, specificity, PPV and NPV of PTX-3 and other CSF biomarkers within the same patient cohort. I would like to bring the Authors´ attention to an excellent study of prof. T. O. Kleine et al. "New and old diagnostic markers of meningitis in cerebrospinal fluid (CSF)" (Brain Res Bull 2003; 61(3): 287-297) where it is stated that "tests with new markers were more laborious, expensive and time-consuming than CSF lactate test" (as well as other classical CSF tests). The same seems true after 20 years, and until now, none of the "new" markers studied in 2003 entered routine practice of CSF laboratories, at least not in the emergency service setting. It seems to me that with PTX-3 measured by an expensive and probably also time-consuming assay would also be quite impracticle, although such pilot studies are always welcomed and may possibly also bring new insights into disease pathophysiology.

Author response: “B” and “C” tables added to Table 4 and text added on page 14, line 244 to line 251 and page 17 line 312 to line 316 and page 19, line 358 to line 359.

CSF cell count and CSF-lactate are well known and good markers for cerebrospinal infections and difficult to compete with for any new marker, which is also the case in this study, as presented in the added B and C in Table 4. Although there are case reports showing bacterial infections with normal cell counts. PTX3 in our cohort has infection marker abilities comparable to these known markers. It is interesting that this a single protein marker related to CRP and most likely locally produced, induced by IL-1 and TNF-alpha, and not IL-6. In this study PTX3 was measured by a time-consuming Elisa test, and for practical reasons, a faster and cheaper assay should be developed for any clinical use of CSF-PTX3. A “quick-CRP” blood test is available and used especially in smaller clinics by general practitioners, and perhaps a fast and cheap measurement for this other pentraxin PTX3 can be made available also. 

Reviewer #2: The author needs to comment further on their case populations to other studies, especially from other areas, to compare the possible relevance of the test in other areas concerning severity, local origin and clinical case differential conditions.

Author response: We agree that an improved description is necessary. Corrections made p. 17 lines 317 to 326, 

Although other biomarkers are discussed, the results as compared to the other test already performed (i.e. CSF analysis reported, CSF cells and differential counts) can be compared in a diagnostic table similar to table 4 and fig3.

Author response: We agree with this need for additional data and explanation. We have added text page 17, line 312 to line 316. “B” and “C” tables added to Table 4. 

Fig 3: Correlation between CSF PTX3 and CSF lactate has been added. 

minor: a short title should be short

Author response: Short title is changed to “Pentraxin 3 as a diagnostic biomarker”.

Other corrections:

Table 1: An error has been spotted in Table 1. All total patients with known pathology was n=145, one patient from encephalitis group was missed in the “All” column. It has been corrected, and does not influence other calculations. 

P. 15 lines 374 to 375. A new sentence has been added.

---

## [Decision Letter · Decision Letter 1]

7 Feb 2023

Pentraxin 3 in the cerebrospinal fluid during central nervous system infections: A retrospective cohort study.

PONE-D-22-24326R1

Dear Dr. Thomsen,

We’re pleased to inform you that your manuscript has been judged scientifically suitable for publication and will be formally accepted for publication once it meets all outstanding technical requirements.

Kind regards,

Francesco Lolli, M.D., Ph.D.

Academic Editor

PLOS ONE

Additional Editor Comments (optional):

a few typos are evidentiated, with suggested correction, by the second referee. To be taken care in the final version

Reviewers' comments:

Reviewer's Responses to Questions

**Comments to the Author**

1. If the authors have adequately addressed your comments raised in a previous round of review and you feel that this manuscript is now acceptable for publication, you may indicate that here to bypass the “Comments to the Author” section, enter your conflict of interest statement in the “Confidential to Editor” section, and submit your "Accept" recommendation.

Reviewer #1: All comments have been addressed

2. Is the manuscript technically sound, and do the data support the conclusions?

Reviewer #1: Yes

3. Has the statistical analysis been performed appropriately and rigorously? 

Reviewer #1: Yes

4. Have the authors made all data underlying the findings in their manuscript fully available?

Reviewer #1: Yes

5. Is the manuscript presented in an intelligible fashion and written in standard English?

Reviewer #1: Yes

6. Review Comments to the Author

Reviewer #1: The Authors have addressed all comments appropriately and carefully revised their manuscript. I have found only a few formal inaccuracies remaining in the revised text that could possibly be corrected in the published version:

Page 5, line 106: "more than" is not necessary before the symbol "more or equal than"

Page 6, row 149: "3.2" instead of "3,2" (decimal point rather than decimal comma, also in Table 1, last row: <3.2)

Page 12-14, Table 4A, B, C, , 1st column under "Microbiologically confirmed pathogen and culture negative samples": "Bacterial" rather than "Bacteriel"

Page 13, Table 4B, 3rd column, line "Microbiologically confirmed and culture negative patient groups": Maximum Youdens J (mmol/L) rather than (pg/ml)

Page 14, Table 4C, 3rd column, line "Microbiologically confirmed and culture negative patient groups": Maximum Youdens J (x10^6/L) rather than (pg/ml)

Page 17, line 314: "slightly better" rather than "slight better"

Page 19, line 358: two commas instead of one after the word "studies"

Finally, I would like to express my gratitude to have the opportunity to review this interesting and carefully written manuscript and to wish the Authors many success in their future research.

7. PLOS authors have the option to publish the peer review history of their article (what does this mean?). If published, this will include your full peer review and any attached files.

Reviewer #1: No

---

## [Editor Report · Acceptance letter]

17 Feb 2023

PONE-D-22-24326R1 

Pentraxin 3 in the cerebrospinal fluid during central nervous system infections: A retrospective cohort study. 

Dear Dr. Thomsen:

I'm pleased to inform you that your manuscript has been deemed suitable for publication in PLOS ONE. Congratulations! Your manuscript is now with our production department. 

Kind regards, 

on behalf of

Dr. Francesco Lolli 

Academic Editor

PLOS ONE